# Pandemic Meets Endemic: The Role of Social Inequalities and Failing Public Health Policies as Drivers of Disparities in COVID-19 Mortality among White, Black, and Hispanic Communities in the United States of America

**DOI:** 10.3390/ijerph192214961

**Published:** 2022-11-14

**Authors:** Lorraine Frisina Doetter, Pasquale G. Frisina, Benedikt Preuß

**Affiliations:** 1Collaborative Research Centre (CRC) 1342 & Research Center on Inequality and Social Policy (SOCIUM), The University of Bremen, 28359 Bremen, Germany; 2University Health Services, Princeton University, Princeton, NJ 08544, USA; 3Research Center on Inequality and Social Policy (SOCIUM), The University of Bremen, 28359 Bremen, Germany

**Keywords:** COVID-19, mortality, social inequalities, public policy, health disparities, United States

## Abstract

The COVID-19 pandemic placed the United States of America (U.S.) under enormous strain, leaving it with higher deaths during the first wave of the outbreak compared to all other advanced economies. Blacks and Hispanics were among those hardest hit by the virus—a fact attributed to enduring problems related to the social determinants of health adversely affecting Communities of Color (CoC). In this study, we ask which distinct factors relating to policy stringency and community vulnerability influenced COVID-19 mortality among Whites, Blacks, and Hispanics during the first year of the pandemic. To address this question, we utilized a mix of correlational and regression analyses. Findings point to the highly divergent impact of public policy and vulnerability on COVID-19 mortality. Specifically, we observed that state-led measures aimed at controlling the spread of the virus only improved mortality for Whites. However, pre-existing social determinants of health (i.e., population density, epidemiological and healthcare system factors) played a significant role in determining COVID-19 outcomes for CoC, even in the face of stringent containment measures by states. This suggests that state-led policy to address present and/or future public health crises need to account for the particular nature of vulnerability affecting Blacks and Hispanics in the U.S.

## 1. Introduction

During the first wave (defined as 1 February to 26 December 2020) of the COVID-19 pandemic, the United States of America (U.S.) was severely affected, having endured a higher number of deaths (per 100,000 of population) compared to all other high-income economies [1]. As an ongoing public health challenge, the virus not only threatens the physical, psycho-social and financial wellbeing of people, but shines a spotlight on longstanding social inequalities emerging at the intersection of health, socio-economic status, and the constructs of race in the U.S. [2,3]. While persistent, racially-based inequalities in COVID-19 mortality were particularly evident prior to the roll out of the first vaccines beginning early in 2021 [4]. In the early stages of the pandemic, mortality amongst Whites was concentrated in the (frail) elderly, whereas for Communities of Color (or CoC, referring to racially and ethnically minoritized groups in the U.S. This includes Native Americans and Asians alongside Blacks and Hispanics. For the purposes of the present study, we focus on the latter two groups which represent the largest minoritized populations in the country and those most impacted by the pandemic (see e.g., [5]))—particularly Blacks and Hispanics (note that Hispanics represent a highly heterogonous group coming from distinct national backgrounds and health characteristics. As discussed by Escarce et al. (2006) [6], sizable differences exist in the morbidity and mortality rates amongst Hispanics making it difficult to generalize. However, given limitations in space and the lack of data available by the CDC on specific Hispanic sub-groups, we take the group as a whole.)—mortality was far more widespread across age-groups [7,8]. These differences have been attributed to enduring problems related to the social determinants of health that disproportionately and adversely affect CoC in the U.S. and which include factors such as job insecurity, crowded living environments, and poorer access to quality health care [5,9,10]. 

In addition to the more chronic social determinants contributing to health disadvantages for Blacks and Hispanics, research also points to the role of specific factors that resulted in higher COVID-19 mortality due to increased exposure to the virus. These mainly involved the inability to practice social distancing owing to higher levels of labour market participation in the essential workforce [11,12,13], as well as the higher concentration of CoC living in crowded urban areas [12,14]. 

While mounting evidence on the disproportionate impact of COVID-19 on Blacks and Hispanics relative to Whites in the U.S. exists, as well as a good deal of scholars that examine the role of the aforementioned social determinants [5,7,12,13], there is far less clarity as to how specific socio-economic factors collided with racial inequalities and public health measures passed during the pandemic to result in such tragic disparities in mortality. Moreover, where research does address this intersection of variables, the results are not always clear or consistent. 

In line with earlier research by Frisina Doetter et al. (2021) [15], social determinants directly associated with increased risk of exposure and death from COVID-19 during the first year of the pandemic have been established as significantly correlated with elevated mortality for Blacks and Hispanics, but not for Whites. Moreover, in the same study, when looking to the competing role of public policies put into place by state governments to protect people, the authors found that the increased stringency of measures benefited only Whites, but were ineffective for Hispanics and potentially detrimental for the Black population [15]. Taken together, the findings pointed to a systematic mortality advantage favoring only White Americans: whether looking to the role of social determinants in isolation or in conjunction with policy stringency, Whites had significantly better chances of surviving the worst consequences of the pandemic. Curiously, this mortality advantage was sustained by Whites even when coming from similar socio-economic backgrounds as their Hispanic and Black counterparts [15].

The racial bias seen in COVID-19 mortality leaves many questions unanswered: How can shared socio-economic characteristics render one societal group more vulnerable to mortality than another? Moreover, how can policy measures applied with similar levels of stringency do a better job of protecting Whites than Blacks and Hispanics from the very same virus? Results from the aforementioned study suggest the need for further research that looks more closely at the nature of vulnerability afflicting CoC. To do so, we revisit earlier research by Frisina Doetter et al. (2021) [15]. This time, however, we unpack the indices used by the authors to measure policy stringency and vulnerability in order to now examine how specific sub-variables contributed to divergent mortality outcomes for Whites, Blacks, and Hispanics during this period. By identifying the social determinants and types of vulnerability that matter most for each group in relation to COVID-19, this research aims to inform and better direct future policy to protect all Americans and contribute to greater health equity in the face of ongoing and newly emerging public health challenges in the U.S.

## 2. Materials and Methods

The research question posed in the present study is: how did distinct factors relating to policy stringency (variable one—independent) and vulnerability (variable two—independent) influence COVID-19 mortality (variable three—dependent) for Whites, Blacks, and Hispanics during the first year of the pandemic? To answer this question, the indices initially used by Frisina Doetter et al. (2021) [15]—hereafter, study one—to measure independent variables one and two were employed once again. This involved returning to the Oxford COVID-19 Government Response Tracker (OxCGRT), which comprises three sub-indices for policy stringency, and the COVID-19 Community Vulnerability Index (CCVI), which encompasses data on seven themes for vulnerability. In what follows, a description of the study’s design and methods is provided, beginning with our period of observation, followed by the measurement of all variables, and, finally, an overview of all statistical testing employed.

### 2.1. Period of Observation

As the present study revisits and deepens the analysis of study one, the same period of observation was adopted: 1 February to 26 December 2020. This ensured the comparability of results between studies and allowed us to concentrate on the effects of policy measures and vulnerability on COVID-19 mortality for each racially-defined group prior to the introduction of mass vaccinations in the U.S. [16].

### 2.2. Policy Stringency

Policy stringency was measured using the OxCGRT, which rates government responses to the pandemic in three sub-indices on a scale from 0 (lowest stringency) to 100 (highest stringency). The three sub-indices of policy stringency with their respective variables/indicators are listed in what follows and can be found in Table 1 below (The indicators for the OxCGRT sub-indices are described at https://github.com/OxCGRT/covid-policy-tracker (accessed on 28 October 2022). The indicators were defined in line with the state-of-the-art on measures to protect people from exposure to the virus, as put forth by leading institutions of public health at the time—i.e., CDC in the U.S. and World Health Organization): Containment and Health Index (14 indicators in total; C1–C8, H1–H3 and H6–H8): closings of schools and universities (C1), workplace closings (C2), cancelling public events (C3), restrictions on gatherings (C4), closing public transport (C5), stay at home requirements (C6), restrictions on internal movement (C7), international travel controls (C8), public information campaigns (H1), testing policy (H2), contact tracing (H3), facial coverings (H6) and vaccination policy (H7).Stringency Index (nine indicators in total; all C indicators plus H1, see above).Economic Support Index (two indicators in total; E1 and E2): Income support (E1), debt/contract relief for households (E2).

It is important to note that, when tested, the sub-indices correlate significantly with one another, affirming the internal validity of the OxCGRT as a measure. In line with the methods of study one and earlier research by Hale et al. (2020) [17], the maximum values for each state and sub-index across the period from 1 January to 21 December 2020 were taken to represent the final scores used in our analysis (Note, because of some small changes in the calculation of the OxCGRT during the last months, there are small differences concerning the values of the OxCGRT in Study 1 and the present contribution. The formulas for the calculation of the sub-indices are described at https://github.com/OxCGRT/covid-policy-tracker/blob/master/documentation/index_methodology.md (accessed on 28 October 2022). It also describes changes in costing that have been made since the data were collected for this analysis).

### 2.3. Vulnerability 

Vulnerability was measured using the CCVI developed by the Surgo Foundation (For more information on the index and Surgo Foundation, see https://precisionforcovid.org/ccvi, accessed on 28 October 2022). The CCVI is a modular index that was built to capture the multi-dimensionality construct of vulnerability toward COVID-19 spanning health, economic, and social disadvantages at the neighborhood level (i.e., census tract). The CCVI is computed using 40 measures across seven themes. The CCVI overall score as well as the seven theme indices range from zero to one, with one representing the most vulnerable geographical area. The seven CCVI themes with their respective variables are listed below:Socioeconomic status (SES): below poverty, unemployed, income, no high school diploma, and percentage of population uninsured.Minority Status and Language: minority status (all persons except white, non-Hispanic) and speaks English less than well.Housing and Transportation Factors: Multi-unit structures, crowding, no vehicle, group quarters, aged 17 or younger, single-parent household, access to indoor plumbing, mobile homes, and older than age 5 with a disability.Epidemiological Factors: Cardiovascular conditions, respiratory conditions, immunocompromised, obesity, diabetes, and aged 65 or older.Healthcare System Factors: Health system capacity, health system strength, healthcare accessibility, and health system preparedness.High Risk Environments: Percentage of population working or living in environments with high infection risk.Population Density: Estimated total number of people per unit area (sq. miles).

The following categories for scores were defined by the Surgo Foundation and also used in the present study: very low (<0.20), low (0.20–0.40), moderate (0.40–0.60), high (0.60–0.80), and very high vulnerability (>0.80 to 1.0).

### 2.4. COVID-19 Mortality

As in study one, we used publicly available de-identified datasets of COVID-19 mortality. Mortality was measured using the standardized death rate (SDR) for COVID-19 related deaths per 100,000 people. The SDR was calculated for three racial groups (i.e., Whites, Blacks, and Hispanics) and is standardized for age to control for the effect of the elderly bias. Accordingly, we can compare the mortality within the racial groups. Therefore, we returned to data from the Centers for Disease Control and Prevention (CDC) for COVID-19 related deaths for Whites, Hispanics and Blacks for each state and population statistics differentiated by age and ethnicity for the year 2019 [11].

### 2.5. Statistical Methods

The present study used the mean and standard deviation as descriptive statistics for all (sub)variables (see Table 2). We also utilized the Pearson’s Product Moment correlations to test the strength of the relationship between all (sub)variables (see Table 3). Stepwise multiple regression analyses (i.e., the stepwise selection method was entered through SPSS v26) were utilized to explore the relationship between chronic social determinants of health (i.e., CCVI) and policy-stringency (i.e., OxCGRT) on mortality rates (standardized by age to control for an elderly bias) among Whites, Blacks, and Hispanics (see Table 4). For the stepwise multiple regression analyses, we ran an adjusted model using “SDR White” as the dependent variable (controlling/standardizing for age) with the seven CCVI themes (i.e., SES, Minority/Language Status, Household/Transportation, Epidemiological, High Risk Environments, Population Density) and the three sub-indices of the OxCGRT (i.e., Containment Health Index, Stringency Index, and Economic Support) serving as independent variables. We ran two other models using “SDR-Black” and “SDR-Hispanic” as the dependent variables of interest (controlling/standardizing for age within each), and the independent variables remained the same. 

## 3. Results

### 3.1. Descriptive Statistics

We began our analysis with descriptive statistics. As seen in Table 2 below, the seven CCVI themes indicated moderate degrees of community vulnerability toward COVID-19 when aggregated across all 50 states. Specifically, the mean CCVI scores ranged between 0.49 to 0.50 which falls within the moderate vulnerability index range as established by the Surgo Foundation (Note for more information, see once again https://precisionforcovid.org/ccvi, accessed on 28 October 2022). Concerning policy stringency, the Containment Health Index demonstrated the lowest average value and the Stringency Index was the highest. The comparison of the SDR of Whites, Blacks, and Hispanics showed significant differences as described in study one [15]. Specifically, the SDR for COVID-19 was highest among Blacks, followed by the Hispanic and White populations.

### 3.2. Correlation Analysis 

The results of the Pearson’s Product Moment correlations can be found in the correlation matrix in Table 3 below. The significant correlations of SDR likely reflect differences in mortality among states independent of racially-defined groups. The CCVI categories show several significant positive and negative correlations. Although significant inter-correlations were observed among several of the independent variables, tolerance statistics did not indicate multicollinearity within our regression analyses.

### 3.3. Regression Analysis

The stepwise regression analysis indicated an overall model of just one significant independent variable (i.e., OxCGRT stringency index) of SDR among Whites, R^2^ = 0.104, R^2^_adj_ = 0.086, F(1, 49) = 5.594, *p* < 0.05. This model accounted for 10.4% of the total variance for SDR within this group. However, the OxCGRT stringency index significantly predicted reduced COVID-19 mortality among the White population (see Table 3). 

A stepwise regression analysis indicated an overall model of two significant independent variables (i.e., CCVI population density and healthcare system factors) for SDR among the Hispanic population, R^2^ = 0.474, R^2^_adj_ = 0.452, F(2, 47) = 21.21, *p* < 0.001. This model accounted for 47.4% of the total variance for SDR within this group. As seen in Table 4, population density (β = 0.561, *p* = 0.000) was a stronger predictor of increased COVID-19 mortality relative to healthcare system factors β = 0.255, *p* = 0.027) for the Hispanic population. 

A stepwise regression analysis indicated an overall model of two significant independent variables (i.e., CCVI population density and epidemiological factors) for SDR among the Black population, R^2^ = 0.373, R^2^_adj_ = 0.347, F(2, 47) = 13.992, *p* < 0.001. This model accounted for 37.3% of the total variance for SDR within this group. As seen in Table 3, population density (β = 0.616, *p* = 0.000) was a stronger predictor of increased COVID-19 mortality relative to epidemiological factors (β = 0.494, *p* = 0.000) for the Black population. 

## 4. Discussion

The results of the present study deepen those of study one conducted by Frisina Doetter et al. (2021) [15]. During the first year of the pandemic, the mortality of Whites benefited from state-led policies to reduce COVID-19 exposure and transmission through measures such as stay-at-home policies and school closures covered by the variable policy stringency measured here (i.e., OxCGRT Index). Moreover, COVID-19-specific vulnerability, captured by the CCVI index, did not play a role in determining the mortality of Whites. While the overall regression model for this group only accounted for 10.4% of variance, what is clear is that Whites were not (nearly as) endangered by the virus based on the social determinants of health compared to their Black and Hispanic counterparts. Instead, the group gained from the types of public health measures taken by states—and where these measures were most stringent, COVID-19 mortality (measured in SDR) for Whites decreased accordingly. 

The same cannot be said for CoC in the U.S.: not only did Blacks and Hispanics suffer disproportionately high mortality relative to their population sizes (see, e.g., [11,15]), but they also failed to see the benefits of the same public policies. This fact is likely underpinned by differences in labor force participation and vulnerability amongst the three groups: the largest share of workers employed in areas not amenable to remote arrangements (especially in production, transportation, material moving, natural resources, construction, and maintenance) is made up by Blacks and Hispanics [18]. These tend to be lower paying jobs, which place workers at risk of poverty and job insecurity, and which—in the context of the pandemic—forced individuals to work and commute despite stay-at-home policies [13,18]. 

Whereas policy stringency did not have a demonstrable effect—positive or negative—on COVID-19 mortality for CoC, the social determinants of health certainly did. For both Blacks and Hispanics, population density played a significant role in driving up mortality, pointing to potential factors such as urban crowding, hygiene, and the inability to practice physical distancing within cities as sources of transmission. While the precise impact of population density on the spread of the virus is not yet clear (see e.g., [19,20]), many studies emphasize its significance either alone or in conjunction with other factors such as metropolitan size and the social and economic connectivity between counties within an area [21,22,23]. For the purposes of the present study, we do not exclude the possible role of these additional factors. Instead, it is important to note that CoC were (and remain) most endangered by the virus, as Blacks and Hispanics disproportionately make up the urban population in the U.S., particularly in the largest metropolitan areas [24]. Moreover, within cities, segregation along racial lines overwhelmingly pushes Blacks and Hispanics into the poorest, most over-crowded and least resourced neighborhoods [25]. This may be the reason why the mortality of Whites, also living in densely populated areas, did not emerge as significant. Crucially, these findings are indicative of the social inequalities expressed geographically in the U.S., not only in terms of the urban-suburban divide, but also in the form of residential segregation or ghettoization of CoC which regularly unfolds [26]. 

In contrast to population density, which had a similar impact on Blacks and Hispanics, the groups differed when it came to the second driver of COVID-19 mortality that emerged: here, findings pointed to the significance of epidemiological factors (i.e., cardiovascular conditions, respiratory conditions, the immunocompromised, obesity, diabetes, and aged 65 or older) for Black Americans, versus healthcare system factors (i.e., health system capacity, health system strength, healthcare accessibility, and health system preparedness) for Hispanics. These drivers, which were not significant for Whites, are highly indicative of the role of longstanding social inequalities afflicting CoC. 

Well before the onset of the pandemic, Black Americans suffered disproportionately higher levels of morbidity and excess mortality owing to the types of conditions included under the variable epidemiological factors here [27,28]. The reasons for this have been mainly attributed to social determinants of health that effect numerous dimensions of life for Black Americans and which include challenges such as inadequate health services (see e.g., [28]), greater exposure to environmental pollutants and hazards (see e.g., [29,30,31]), discriminatory practices in education (see e.g., [32]) and housing (see e.g., [33]), as well as in policing/imprisonment (see e.g., [34,35]). The common thread amongst all these issues is the role played by systemic racism, which has been referred to as a social determinant in its own right affecting all areas of health and well-being for Black Americans and ultimately contributing to lower life expectancies (by 4 years) compared to Whites [34,36,37]. 

In the case of Hispanics, factors rooted in the healthcare system rather than epidemiology emerged as a second major driver of COVID-19 mortality. Interestingly, despite coming from lower socio-economic backgrounds, this group generally enjoys greater longevity than even White Americans—a phenomenon often referred to as the ‘Hispanic Epidemiological Paradox’ [38]. This has been attributed to a number of factors such as the tendency for the fittest to migrate; lower rates of smoking; as well as better social networks (see e.g., [38,39,40]). At the same time, however, foreign-born U.S. Hispanics tend to live longer with disability than Whites, with disability rates increasing steeply and disproportionately amongst Hispanic men from the age of 50 [41]. This likely reflects differences in employment, with Hispanics more likely to engage in more physically-arduous labor than other groups [42]. Moreover, research suggests that Hispanic migrants may return to their home country upon getting seriously ill, thereby inflating the longevity of the remaining population in the U.S. [43].

For the purposes of the present study, questions surrounding the overall mortality and increased longevity of Hispanics relative to other groups cannot be adequately addressed. Of interest here is the issue of COVID-19 mortality, for which this population, similar to Black Americans, suffered disproportionately higher death rates compared to Whites. This means that the pandemic hit Hispanics so heavily that it even offset the presumed health advantages that might otherwise contribute to increased longevity. Crucially, healthcare system factors—the second driver of mortality found here—may be key to understanding these developments. For a group elsewhere reported (see e.g., [9,10,25,44]) as systematically excluded or disadvantaged by the American healthcare system, the pandemic presented a situation in which timely access to quality health services was imperative but also elusive for many [45]. This, coupled with the aforementioned role of population density, the enormous influx of Hispanics into the largest metropolitan areas over the last decades [25], and the potential role of irregular migration/undocumented workers [44,45] likely coalesced to increase the exposure of Hispanics to the virus, and, once infected, led them to receive poorer quality care and/or to avoid the take-up of services altogether. The latter is of particular significance given the fact that Hispanics are three times more likely than Whites to live without health insurance in the U.S. [41,45].

This study has several limitations, largely due to limitations in the available data. First, we were unable to assess the implications of policy on COVID-19 mortality by stratifying data by U.S. geographical region (e.g., urban vs. rural counties). However, there is evidence that CoC are at increased risk for COVID-19 mortality within rural and urban environments because of similar social vulnerabilities similarly captured by the CCVI from our study (e.g., socioeconomic status and housing status). Another limitation of the current study is that we were unable to examine the relative role of sex/gender on COVID-19 mortality because such data is not available at the federal (i.e., CDC) level in the U.S. Recent studies have attempted these analyses but had to utilize more local/state level data by comparing mortality within a subsample of states (i.e., Georgia vs. Michigan), and found differences in COVID-19 mortality by sex and race [46,47]. Moreover, the Kaiser Family Foundation Coronavirus and Health Tracking Polls reported that men were less likely to report taking protective actions toward COVID-19 as recommended by governmental advisories [48]. Although behavioral differences regarding adherence to governmental advisories could account for mortality differences by sex, they do not address the racial disparities on COVID-19 mortality observed in the current study. For these reasons, it would be important for future research to explore the intersectionality of sex, race, geographical region, and governmental policy adherence to prevent and contain the spread of COVID-19 [49]. 

Thus far, owing to a paucity of research on the subject, it is still unclear as to the exact pathways that led from exposure, to transmission/infection, to care, and then resulted in the disproportionate mortality experienced by Hispanics during the pandemic [44]. The present findings can only speak to the overall drivers of mortality for each group included in the study, but cannot trace how the underlying mechanisms played out within and between them. What is clear, however, is that the social determinants of health specific to COVID-19-vulnerability that were covered by the CCVI did not impact unfavourably upon Whites, but had an enormous impact on CoC. Furthermore, this impact could not be softened by the types of public policies passed at the time, which did not attend to the socio-economic and the pernicious long-term effects of systemic racism in the U.S.

## 5. Conclusions

Taken together, the present study establishes that greater stringency of state-led measures aimed at controlling the spread of the virus does not lead to improved mortality rates for CoC and Whites equally. Regrettably, the role of pre-existing social determinants of health played a significant part in determining COVID-19 outcomes observed for Blacks and Hispanics, even in the face of rigorous measures taken by states. What is made abundantly clear by the present findings is that policymaking passed during a crisis cannot subdue the impact of longstanding risks and conditions associated with race/ethnicity-based social inequalities in the U.S. Moreover, given these inequalities, a one-size-fits-all approach to combating future public health crises cannot work to protect vulnerable populations. What is needed instead is far greater attention to the socio-economic realities of CoC during the short-term, as well as a much broader vision of improving conditions in the long-term in order to ensure well-being above and beyond the context of the COVID-19 pandemic. In line with our findings, the overall health and well-being of Black Americans needs to be addressed in order to tackle the types of morbidities suffered by this population which rendered them especially vulnerable to the most pernicious consequences of COVID-19. In the case of Hispanics, this means dismantling the structural barriers to timely access to quality healthcare adversely affecting this population. Finally, one major form of social inequality that needs addressing for long-term well-being concerns what appears to be the universal impact of geography on CoC. Population density was a major driver of COVID-19 mortality in the current study. However, population density should be viewed as a surrogate variable for many other forms of social inequalities encompassed within COC—namely, the increased burden of chronic disease risk factors, healthcare access barriers, limited access to healthy foods, the need to work or else, and the inability to practice social distancing [45]. Addressing the multiple determinants of health both at the governmental-policy and healthcare system levels can advance health equity within the U.S., especially for groups historically linked to systematic discrimination or exclusion [45,50].

## Figures and Tables

**Table 1 ijerph-19-14961-t001:** Overview of OxCGRT Indicators.

Indicators		Index Name		
ID	Name	Containment and Health Index	Stringency Index	Economic Support Index
C1	School closing	x	x	
C2	Workplace closings	x	x	
C3	Cancel public events	x	x	
C4	Restrictions on gatherings	x	x	
C5	Close public transport	x	x	
C6	Stay at home requirements	x	x	
C7	Restrictions on internal movement	x	x	
C8	Restriction on international travel	x	x	
E1	Income support			x
E2	Debt/contract relief for households			x
H1	Public information campaigns	x	x	
H2	Testing policy	x		
H3	Contact tracing	x		
H6	Facial coverings	x		
H7	Vaccination policy	x		
H8	Protection of elderly people	x		

x = indicator is included in the index.

**Table 2 ijerph-19-14961-t002:** Description of the seven categories of the CCVI, the three sub-indices of the OxCGRT and the SDR for each group.

		N	Mean	Std. Deviation
		Valid		
Mortality (SDR)	SDR White	50	52.10	26.93
	SDR Black	50	103.00	83.40
	SDR Hispanic	50	97.56	73.74
Vulnerability (CCVI)	Socioeconomic status	50	0.50	0.30
	Minority Status & Language	50	0.49	0.30
	Household & Transportation	50	0.50	0.30
	Epidemiological Factor	50	0.50	0.30
	Healthcare System Factors	50	0.51	0.29
	High Risk Environments	50	0.51	0.29
	Population Density	50	0.49	0.29
Policy stringency (OxCGRT)	Containment Health Index	50	69.46	5.85
	Stringency Index	50	80.03	6.13
	Economic Support Index	50	77.25	12.55

**Table 3 ijerph-19-14961-t003:** Correlation matrix of the seven categories of the CCVI, the three sub-indices of the OxCGRT and the SDR for each group.

Pearson r	SES	Minority Status & Language	Household & Transportation	Epidemiological Factor	Healthcare System Factors	High Risk Environments	Population Density	Containment Health Index	Stringency Index	Economic Support Index
SDR White	0.09	0.09	−0.16	0.25	0.12	0.15	0.19	−0.10	−0.32 *	0.10
SDR Black	0.14	0.27	−0.35 *	0.24	0.29 *	0.12	0.41 **	0.17	−0.06	0.27
SDR Hispanic	0.11	0.50 ***	−0.45 **	−0.16	0.44 **	0.06	0.65 ***	0.08	−0.07	0.13
SES	1	0.43 **	0.47 ***	0.35 *	0.49 ***	0.06	−0.01	0.10	0.12	0.06
Minority Status & Language		1	−0.33 *	−0.35 *	0.45 ***	−0.34 *	0.76 ***	0.28 *	0.11	0.09
Household & Transportation			1	0.34 *	0.003	0.16	−0.70 ***	−0.19	−0.06	−0.21
Epidemiological Factor				1	−0.003	0.27	−0.41 **	0.01	0.05	0.02
Healthcare System Factors					1	0.33 *	0.33 *	−0.20	−0.23	−0.16
High Risk Environments						1	−0.24	−0.39 **	−0.26	−0.25
Population Density							1	0.21	0.06	0.14

*** correlation is significant at the 0.001 level (two-tailed), ** correlation is significant at the 0.01 level (two-tailed), * correlation is significant at the 0.05 level (two-tailed).

**Table 4 ijerph-19-14961-t004:** Coefficients between each independent variable and SDR for White, Hispanic, and Black Populations.

Dependent Variable	Independent Variable	Unstandardized β	Standardized Coefficient β	t	Sig.
SDR per 100,000 population White	OxCGRT stringency index	−1.419	−0.323	−2.365	0.022
SDR per 100,000 population Hispanic	Population Density	141.993	0.561	5.015	0.000
	Healthcare System Factors	64.431	0.255	2.28	0.027
SDR per 100,000 population Black	Population Density	176.294	0.616	4.863	0.000
	Epidemiological Factors	138.025	0.494	3.900	0.000

## Data Availability

Data used for this study can be found online at the following addresses: For policy stringency measured by the OxCGRT Index, see https://github.com/OxCGRT/covid-policy-tracker (accessed on 28 October 2022). For data on vulnerability to COVID-19 mortality, see https://precisionforcovid.org/ccvi (accessed on 28 October 2022).

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
