# Peer review of "Pandemic Meets Endemic: The Role of Social Inequalities and Failing Public Health Policies as Drivers of Disparities in COVID-19 Mortality among White, Black, and Hispanic Communities in the United States of America"

_ijerph, 2022, doi:10.3390/ijerph192214961_

Round 1

Reviewer 1 Report

This study examined if factors related to policy stringency and vulnerability could influence COVID-19 mortality among White, Black, and Hispanic population in the US. Authors found that more stringent state-level policy only reduced mortality for White population, while pre-existing social determinants had larger influence on mortality for Black and Hispanic population. I have the following comments regarding the data and method:

1.      More description of the data would be helpful. Did authors use administrative data, or survey data? What’s the number of observations? Is there any sample selection, or weighting process involved?

2.      If my understanding is correct, authors used data aggregated at state level. It is possible to use more granular data (e.g. at metropolitan statistical area/MSA level, or county level)? Ideally, if authors could explore variations in mortality among MSAs or counties located on state boarders with similar socio-economic-demographic characters but the states had different level of policy stringency. Authors could explore a regression discontinuity style study design to make a stronger causal argument. Also, within a state, I guess policy stringency could have heterogenous impact across different communities (e.g. urban vs rural, higher vs lower income, communities with more White, Black, or Hispanic population). I feel these are the remaining questions as we keep thinking about this issue.  

3.      Is the regression weighted by state-level population to adjust for the difference between larger (e.g. California, Texas) and smaller (e.g. Rhode Island) states?  

4.      The standardized death rate (SDR) for COVID-19 is age adjusted, is it also gender adjusted? I think gender could be a potential confounder and should be addressed.

Reviewer 2 Report

A lot of work has been done in this area to attribute COVID-19 to social determinants of health, particularly to race, and it is unclear to me how the findings from this manuscript can contribute to the existing literature or expand our knowledge base. The state regulations does not cover a background of discrepancies across the US in terms of what states determined were best public health guidelines. Limitations and implications to this study are lacking. 

Reviewer 3 Report

Well written work. Two changes would improve the study in my eyes - one is to reference more research than the "study one" this manuscript expands on; in other words, expand the literature review by examining what other work has been published in the last two years to address similar questions to yours. This has been done (increasingly). This would enrich and expand your current study by connecting it to other relevant discussions; and not just read as a follow-up.

Second, also for the reader's sake, say more how this study is different than "study one." Expand on the differences, especially the contribution this study seeks to make - one or two sentences before the methods section would suffice. Just add a bit more impact.

On methodology - as a reader a bit outside your discipline, I am not clear why the independent variable "policy stringency" "double-counts" indicators - for example, C1-8 seems counted for both containment and stringency. Why? How does this not lead to more frequent noting of policy stringency than necessary? Same with some of the H criteria. Again, this may be a non-issue methodologically, but it's not clear for non-specialized readers.

Please note some font differences on page 3.

Perhaps statement this study "confirms" the results of "study one" is not necessary (obviously, same data, even some of the same variables. Stating "deepen" is enough.

I look forward to seeing this published, good work, authors.

Round 2

Reviewer 1 Report

This version is better than the last one. Authors addressed almost all of my comments in the limitation section. I understand that, although some additional effort (e.g. additional analysis, discussion) could be helpful.

There is a typo in line 333, should be "reported" not "reprted".

Author Response

Thank you for pointing out the typo.

We will consider your input for future, follow up papers in which we would have more space to conduct additional analysis.